# Research Advances in *Toona sinensis*, a Traditional Chinese Medicinal Plant and Popular Vegetable in China

**Qian Zhao** [1,2], **Xiu-Lai Zhong** [1], **Shun-Hua Zhu** [1], **Kun Wang** [1], **Guo-Fei Tan** [1,3,*], **Ping-Hong Meng** [1,*] **and Jian Zhang** [3,*]

1  Institute of Horticulture, Guizhou Academy of Agricultural Sciences, Guiyang 550006, China; zhaoq217@163.com (Q.Z.); gzzyzxl@foxmail.com (X.-L.Z.); zsh2801@163.com (S.-H.Z.); wangkunfun@hotmail.com (K.W.)
2  Key Laboratory of Plant Resource Conservation and Germplasm Innovation in Mountainous Region (Ministry of Education), Collaborative Innovation Center for Mountain Ecology & Agro-Bioengineering (CICMEAB), College of Life Sciences/Institute of Agro-Bioengineering, Guizhou University, Guiyang 550025, China
3  Faculty of Agronomy, Jilin Agricultural University, Changchun 131018, China
*  Correspondence: tagfei@foxmail.com (G.-F.T.); mengph322@163.com (P.-H.M.); jian.zhang@ubc.ca (J.Z.)

**Abstract:** *Toona sinensis*, a perennial and deciduous tree belonging to the Meliaceae family, has been cultivated for more than 2000 years in China. Storing the buds of *T. sinensis* is difficult, as it is easy for them to rot during storage, which seriously affects their edible and commodity value. Young leaves and buds of *T. sinensis* plants are excellent source of flavonoids, terpenoids, phenylpropanoids, and more. In addition, the bioactive components of *T. sinensis* possess numerous health benefits, such as antiviral, antioxidant, anti-cancer, anti-inflammatory, and hypoglycemic effects. In this review, we summarize the storage and preservation, nutritional components, specific chemical compounds, pharmacological value, function genes, and omics of *T. sinensis*. This review aims to provide basic knowledge for subsequent future research seeking to understand the comprehensive biology and use of *T. sinensis* as a favored Chinese food and pharmacological resource.

**Keywords:** *T. sinensis*; preservation; nutritional components; pharmacology; functional gene; omics





## 1. Introduction

*Toona sinensis* (A. Juss.) Roem (Figure 1) belongs to the Meliaceae family and is commonly called Chinese toon or Chinese mahogany. It is a woody perennial deciduous tree [1]. *T. sinensis* has a cultivation history stretching back more than 2000 years in China [2]; it is widely distributed throughout China, ranging from Liaoning in the east to Gansu in the west, Guangdong, Guangxi, and Yunnan in the south, and southern Inner Mongolia in the north. Of all the provinces, Anhui, Shandong, Henan, and Hebei have the most areas under *T. sinensis* cultivation [3,4]. *T. sinensis* is a medicinal and edible vegetable, and different tissues of this plant have been used to treat a wide variety of diseases [5,6]. Phytochemical investigations of *T. sinensis* have showed that its main constituents include terpenoids, phenylpropanoids, and flavonoids [7–9], and that the plant has many pharmacological activities, including anti-tumor, anti-oxidant, anti-diabetic, anti-inflammatory, antibacterial, and anti-virus action [10–12].

*T. sinensis* is divided into two types based on the color of tender leaves and petioles: red *T. sinensis* and green *T. sinensis* [13,14]. Red *T. sinensis* is rich in anthocyanins, thus the leaves and petioles are purple [15]. Red *T. sinensis* is more popular with consumers. This plant has a short harvest period, and storage is difficult; the edibility and commodity value is lost after 2–3 days at normal temperature [16].

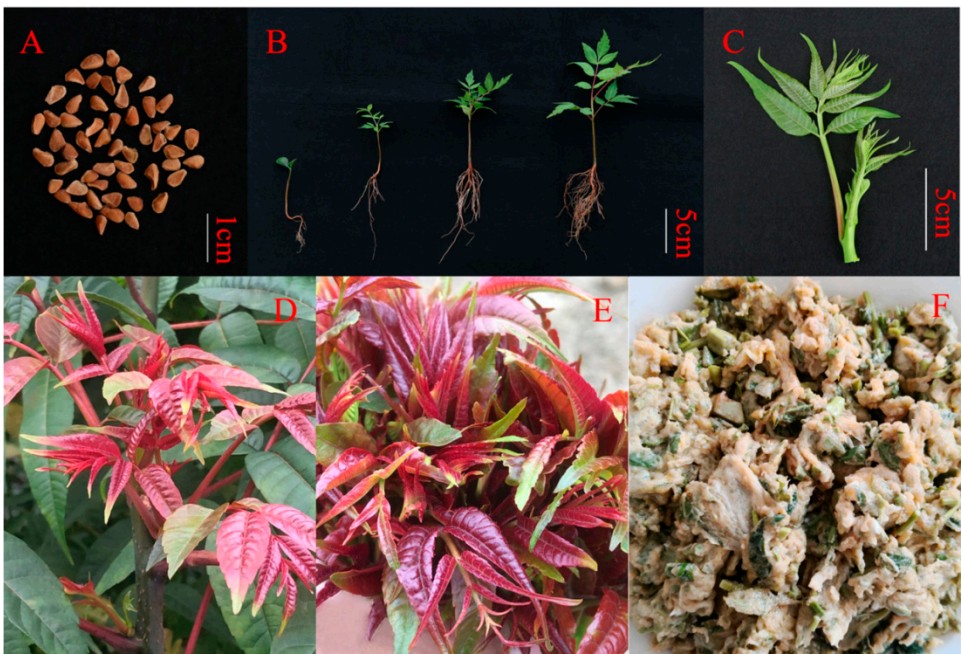

**Figure 1.** Seeds (**A**), seedlings (**B**), plant and buds (**C–E**) of *T. sinensis*, and cooked with eggs (**F**).

Currently, *T. sinensis* has aroused considerable public interest in terms of storage and preservation, nutritional components, medicinal use, and gene and omics research. However, there has been no systemic review of these aspects. Consequently, this paper aims to summarize the current advances in storage and preservation, nutritional components, medicinal use, and gene and omics studies of *T. sinensis*. Furthermore, this paper discusses potential future development perspectives on *T. sinensis*.

## 2. Storage and Preservation of *T. sinensis* Buds

Storage environmental conditions, including temperature [17], humidity [18], gas composition [19], etc., have been shown to affect the quality of the *T. sinensis* buds. Currently, researchers often use physical, chemical, and biological means to store and preserve *T. sinensis* buds. Although the principles of the three means are different, they all delay the duration and rot process of *T. sinensis* by controlling the water evaporation, respiration, and relative humidity of the environment of this plant and improve preservation time and quality of *T. sinensis* buds.

### 2.1. Physical Preservation Method

#### 2.1.1. Temperature Control Preservation Technology

Temperature control preservation technology mainly includes low temperature preservation, low temperature and high humidity preservation, and quick-frozen preservation. Low temperature preservation can delay the metabolic process of *T. sinensis* after harvest; the buds of *T. sinensis* are packed into a plastic bag and stored at 0 °C, and can be stored for up to two months [20]. The low temperature and high humidity preservation method can slow down the loss of nutrients in the vegetables; this method can effectively inhibit the activity of polyphenol oxidase and maintain the quality of the plant [21,22]. Quick-frozen preservation can maximally maintain the natural quality of *T. sinensis* buds, which could be stored for about a year using this method [23].

#### 2.1.2. Dehydration Preservation Technology

Fresh buds of *T. sinensis* contain up to 90% water, and the leaves are prone to wilting after picking. Dehydration preservation technology reduces the water content of vegetables to keep the color and nutrient content of vegetables unchanged [24]. Zhao et al. (2014) used

the methods of hot drying, microwave drying, vacuum drying, and vacuum freeze drying to treat *T. sinensis*, among which vacuum freeze drying best maintained the nutritional composition of the plant [25]; the sensory quality was better and the rehydration rate reached 51.05%. Lv (2010) conducted research about vacuum freeze-drying of *T. sinensis* buds. The products were obviously better than those obtained by pickled storage and ordinary refrigeration; the buds of *T. sinensis* had a shelf life of up to two years and were essentially the same in terms of nutrient content [26].

### 2.1.3. Modified Atmosphere Preservation Technology

Modified atmosphere preservation is used to adjust the proportion of gases in the storage environment, in order to inhibit the respiration and evaporation of fruits and vegetables [27,28]. Zhu et al. (2014) used two types of modified atmosphere packaging materials with difference thickness (0.01 and 0.03 mm), namely, low density polyethylene (LDPE) and high density polyethylene (HDPE), on *T. sinensis* storage, and the experimental results showed that the packaging material of low-density polyethylene (thickness 0.03 mm) effectively inhibited the defoliation phenomenon of *T. sinensis* buds and delayed the aging of *T. sinensis* [29].

### *2.2. Chemical Preservation Method*

Chemical preservation means are mainly used to extend the preservation time of buds of *T. sinensis* by adding chemical preservatives. Although the preservation effect is good, it may be harmful to the human body and the environment. There are three main types of preservatives: adsorption, anti-corrosion, and inhibition. Ethylene adsorbent is often used in the storage and preservation of *T. sinensis*. It can delay post-ripening and achieve the preservation effect of buds of *T. sinensis* [30]. Preservatives mainly use carbendazim to kill pathogenic microorganisms on the surface of *T. sinensis* buds to attain the effect of preventing *T. sinensis* bud rot [31]. 6-Benzylaminopurine is a common inhibitory preservative in *T. sinensis* preservation which can prevent the aging of *T. sinensis*. Results show that after 45 days of storage, the color and fragrance of *T. sinensis* are basically unchanged [32].

### *2.3. Biological Preservation Method*

Bio-preservatives are natural substances extracted from plants or animals. With the advantages of low pollution and low cost, bio-preservation technology is widely used in the market [33]. Chen et al. (2015) treated *T. sinensis* with 8 mmol·L$^{-1}$ exogenous betaine; the resulting rotting of *T. sinensis* buds after sixteen days of storage was not serious, the loss of chlorophyll, Vc, and total flavonoid content was alleviated, and the plant retained its commercial value [34]. Zhang et al. (2009) treated *T. sinensis* buds with *Allium macrostemon* Bunge extracts; the results showed that *A. macrostemon* Bunge extracts could significantly reduce the decay rate of *T. sinensis* and inhibit the decline of Vc content while extending the shelf life of *T. sinensis* buds by more than seven days [35].

## 3. Eating and Processing of *T. sinensis*

### *3.1. Cooking Methods*

There are three traditional ways to eat *T. sinensis* buds: (1) fresh *T. sinensis* buds; (2) old dish *T. sinensis* buds; and (3) fried *T. sinensis* buds. Fried *T. sinensis* buds are mainly eaten with flour or lotus root. *T. sinensis* buds can be wrapped directly with flour paste, or the *T. sinensis* buds can be chopped and used to fill the holes of lotus root, then wrapped with flour paste and fried with oil before sprinkling them with spices. These edible methods are very popular and taste very delicious.

### *3.2. Processing of T. sinensis Buds*

In recent years, with people obtaining a deeper understanding of the health value of *T. sinensis*, more and more ways to eat *T. sinensis* have been developed. In addition to

traditional eating methods, this plant is often processed into various products (Table 1). There are no studies reporting on *T. sinensis* bud preservation using radiation exposure, which could be a safe and low-cost procedure.

**Table 1.** Processed products using *T. sinensis*.

| Products | Reference |
|---|---|
| Natural *T. sinensis* powder | [36] |
| Red *T. sinensis* lactone tofu | [37] |
| Liquid flavoring agent | [38] |
| Canned *T. sinensis* | [39] |
| *T. sinensis* bread | [40] |
| *T. sinensis* noodles | [41] |
| *T. sinensis* biscuit | [42] |
| *T. sinensis* Hotpot condiment | [43] |
| *T. sinensis* seasoning oil | [44] |
| *T. sinensis* juice | [45] |
| *T. sinensis* coffee ice-cream | [46] |
| *T. sinensis* yogurt | [47,48] |

## 4. Nutrient Composition of *T. sinensis*

*T. sinensis* is an excellent source of minerals, proteins, carbohydrates, fatty acids, amino acids, carotene, vitamins, dietary fiber and other compounds [49,50]. The nutritional components of different tissues and varieties of the plant are different.

Jin and Dong (1994) analyzed the content of amino acids, soluble sugars, and fatty acids in *T. sinensis* tender buds (Table 2); the results showed that the essential amino acid and total amino acid content accounted for 8.51% and 27.43% of dry weight (DW), respectively. Gas chromatography (GC) detected 22.05% palmitic acid, 1.88% stearic acid, 3.44% oleic acid, 30.16% linoleic acid, and 42.47% linolenic acid. The older leaves of *T. sinensis* are rich in nutrient composition as well, and yield more than the young buds [51]. However, the old leaves are often discarded. Hou et al. (1997) analyzed the protein, amino acid (Table 3), fat, crude fiber, and mineral content (Table 4) of old leaves, and found that their crude protein, amino acid, and fat contents were richer than those of corn, showing the old leaves of *T. sinensis* to be a better animal feed [52]. Xu et al. adopted the full-distance equal division method to rank different varieties based on their content of nutritional compositions, resulting in the following ranking: Red Chinese toon > Red-leaved Chinese toon > Sprout Chinese toon > Black-oil Chinese toon > Red-oil Chinese toon > Brown Chinese toon > Green-oil Chinese toon > Green Chinese toon > Peach Chinese toon [3].

**Table 2.** Amino acids found in *T. sinensis* buds.

| Amino Acid | Content (%) | Amino Acid | Content (%) |
|---|---|---|---|
| Threonine * | 1.04 | Aspartate | 2.29 |
| Valine * | 1.39 | Serine | 1.28 |
| Methionine * | 0.58 | Glutamic acid | 7.24 |
| Leucine * | 1.13 | Proline | 2.75 |
| Isoleucine * | 1.97 | Glycine | 1.44 |
| Phenylalanine * | 1.19 | Alanine | 1.49 * |
| Lysine * | 1.21 | Cystine | — |
| Tryptophan * | — | Tyrosine | 0.89 |
| Arginine | 1.15 | Histidine | 0.39 |

Note: * Represents essential amino acids.

**Table 3.** Amino acids found in old leaves of *T. sinensis*.

| Amino Acid | Content (%) | Amino Acid | Content (%) |
|---|---|---|---|
| Threonine * | 0.74 | Aspartate | 1.11 |
| Valine * | 0.83 | Serine | 0.80 |
| Methionine * | 0.30 | Glutamic acid | 2.60 |
| Leucine * | 1.36 | Proline | 0.73 |
| Isoleucine * | 0.72 | Glycine | 0.80 |
| Phenylalanine * | 0.86 | Alanine | 0.94 |
| Lysine * | 0.66 | Cystine | 0.04 |
| Tryptophan * | — | Tyrosine | 0.68 |
| Arginine | 0.89 | Histidine | 0.16 |

Note: * Represents essential amino acids.

**Table 4.** The content of main mineral elements in old leaves of *T. sinensis* (content per gram).

| Zn (mg) | Cu (mg) | Mn (mg) | Na (mg) | Fe (mg) | Mg (mg) | K (mg) | Ca (mg) |
|---|---|---|---|---|---|---|---|
| 0.059 | 0.022 | 0.047 | 0.807 | 0.187 | 3.504 | 20.300 | 5.931 |

Many researchers have reported that the content of nutritional composition in different provenances and harvest periods is different. For instance, Yang et al. investigated the changes of nutritional composition at four different harvest periods from six different provenances [53], and Wang et al. analyzed the nutritional composition at five different harvest periods from five different provenances [54]. Both studies found significant differences in content of amino acids, proteins, soluble sugars, and vitamin C in *T. sinensis* at the same period from different provenances or at different harvest periods from the same provenance.

## 5. Characteristic Chemical Compounds of *T. sinensis*

### 5.1. Volatile Compounds

The headspace solid phase microextraction (HS-SPME) and gas chromatography-mass spectrometry (GC-MS) technologies can be used to extract and analyze the volatile components of plants. Yang et al. (2016) used an ultrasonic-assisted method to extract volatile components from *T. sinensis* leaves and identified 73 volatile compounds using GC-MS. The main volatile compounds included arachidonic acid ethyl ester, benzothiazole, pentadecanoic acid methyl ester, n-heneicosane, *β*-caryophyllene, benzoic acid hexylester, 1, 2-benzenedicarboxylic acid butyl octyl ester, limonene, heptacosane, n-hexadecanoic acid, and others [55]. Ji et al. (2018) used HS-SPME and GC-MS techniques to identify 32 volatile compounds from *T. sinensis* leaves, and found that (3E)-3-Hexenyl acetate, (Z)-Hex-3-en-1-ol, caryophyllene, and (Z)-Butanoic acid-3-hexenyl ester were the major constituents [56]. Gao et al. (2016) used the same methods to identify volatile components in *T. sinensis* leaves, flowers, and seeds, and further analysis showed 36 volatile compounds in leaves, 37 volatile compounds in flowers, and 26 volatile compounds in seeds. Among them, beta-Elemene, germacrene B, L-calamenene, and alpha-Cubebene were the most common [57]. Comparing the volatile components of three *T. sinensis* varieties (Ximu red from Yantai city in Shandong province, Jiaozuo red from Jiaozuo city in Henan province, and Heiyou purple from Taihe county in Anhui province), Liu et al. (2013) discovered that the Ximu and Jiaozuo red cultivars contain thiophenes and terpenes, although their exact contents differ, and the Heiyou purple cultivar contains terpenes and esters [58].

### 5.2. Terpenoid Compounds

After the first terpenoid compound, toosendanin, was extracted in 1972, ongoing research has identified various terpenoids in *T. sinensis* leaves; triterpenoids are main type of terpenoid. Hu et al. (2020) used various chromatographic techniques (e.g., silica gel, Sephadex LH-20, MCI gel, and ODS gel) to isolate and characterize terpenoids

from 80% ethanol extract of *T. sinensis* leaves; the results showed that eight terpenoids could be extracted and identified from the leaf extract of *T. sinensis*: cedrelone, cedrodorol B, toonayunnanin D, toonaciliatone D, toonaciliatone A, 8$\beta$-hydroxypimar-15-en-19-oic acid methyl ester, 11$\beta$-acetoxyobacunol, and 11$\beta$-hydroxygedunin [59]. Yang et al. (2013) extracted the triterpenoids betulinic acid and ursolic acid for the first time [60]. Two triterpenoids, 6-acetoxyobaconicate and 7$\alpha$-actoxydihydron, were first isolated and identified from the leaves of *T. sinensis* [61]. Moreover, another study first extracted and identified the terpenoids 11$\beta$-hydroxy-7$\alpha$-oba-conylacetate and 11$\beta$-oxocneorin G from *T. sinensis* leaves [62].

### 5.3. Flavonoid Compounds

Flavonoids are widely distributed in plants, and the flavonoid content in *T. sinensis* leaves is 2~3 times than that of *Ginkgo biloba* L.. Zhao et al. (2016) used high-performance liquid chromatography (HPLC) to extract four flavonoids (rutin, quercetin, kaempferol, and gallic acid) from old leaves of *T. sinensis* [63]. Chen et al. (2019) separated and purified extracts from old leaves using column chromatography and HPLC, then used nuclear magnetic resonance (NMR) and infrared spectroscopy (IR) to identify substances such as rutin, epicatechin, quercetin, isoquercetin, and guava glucoside [64]. Ge et al. (2017) used ultra-high performance liquid chromatography to determine the flavonoid compounds in *T. sinensis* shoots, ultimately detecting glycosides, rutinoside, myricitrin, hyperoside, isoquercitrin, guaijaverin, astragalin, quercitrin, and afzelin [65]. Moreover, Miao et al. (2016) found five flavonoids identified from the *T. sinensis* leaf extract using the NKA9 macroporous adsorption resin method [66]. The total flavonoid content in stems, leaves, and flowers of *T. sinensis* plants were comparatively analyzed, and the results showed that the total flavonoid content in the leaves was highest, followed by the flowers and stems [67].

### 5.4. Phenylpropanoid Compounds

Phenylpropanoids commonly exist in natural plants, and mainly include lignins and coumarins. Nine phenylpropanoid compounds have been isolated and identified from different tissues of *T. sinensis*, namely, cedralins A and B [68], lyoniresinol, toonin C (Figure 2), matairesinol [6], 7-dimethoxy-5-methylcoumarin [69], scopoletin [70], and ficusesquilignans A and B [71]. Phenylpropanoid compounds often have pharmacological activities, such as antiviral, anti-inflammatory, antitumor, and antibacterial activity.

**Figure 2.** Structure of toonin C.

## 6. Pharmacological Characteristics of *T. sinensis*

The medicinal use of *T. sinensis* was first recorded in the Tang Materia Medica, which is a famous Traditional Chinese Medicine (TCM) monograph written in Tang dynasty China; this plant has thus been used as an herbal medicine for thousands of years [72,73]. The "Compendium of Materia Medica" and "Dictionary of Traditional Chinese Medicine" introduced the medicinal uses of the roots, bark, petioles, leaves, fruits, and seeds of *T. sinensis* [74,75]. In Chinese folk medicine, *T. sinensis* is described as an herbal medicine with good anti-inflammatory, detoxifying, and hemostatic effects, and it was commonly used to treat enteritis, dysentery, urinary tract infections, leukorrheal diseases, and skin itch [76]. Modern studies of *T. sinensis* have mainly focused on the extraction and identification of bioactive components from the leaves of *T. sinensis* [77,78], while few studies have reported the bioactive ingredients in the bark and seeds [79]. Extensive studies have shown that bioactive components from *T. sinensis* possess numerous health benefits, such as antiviral, antibacterial, antioxidant, anti-cancer, anti-inflammatory, and hypoglycemic effects [80–84].

### 6.1. Antioxidant Effect

Previous research reports have indicated that the extracts of *T. sinensis* are natural antioxidant agents [85,86]. Several reports have shown that phenolic compounds in the extract of *T. sinensis* have the ability to scavenge free radicals [87–89]. In addition, Hsieh et al. (2004) reported that extract of *T. sinensis* has antioxidant effects on hydrogen peroxide-induced oxidative stress [90].

### 6.2. Antiviral and Antibacterial Effect

*T. sinensis* possess notable antiviral and antibacterial effects. Chen et al. found that extract of *T. sinensis* leaves had antiviral activity against SARS-CoV in vitro, with an IC50 value of 30 μg· mL$^{-1}$ [91]. You et al. (2013) reported that extract of *T. sinensis* leaves could be used an alternative treatment and prophylaxis against the H1N1 virus [92]. In addition, the extract of *T. sinensis* leaves has been found to possess promising antibacterial potential against *E. coli*, *Salmonella*, and *Staphylococcus* [93,94]. At present, the antiviral and antibacterial effects of *T. sinensis* are an increasing concerned of the pharmaceutical industry.

### 6.3. Anti-Inflammatory Effect

Many natural anti-inflammatory products isolated form the extracts of *T. sinensis* have been reported, and play important roles in preventing and treating inflammatory disease [95]. In 2012, Yang and Chen (2012) published a research report showing that total polyphenols extracted from the seeds of *T. sinensis* had a significant effect on the treatment of rat arthritis [96]. Chen et al. (2017) reported 7-deacetylgedunin (7-DGD) extracted from the fruit of *T. sinensis*, conducted in vivo and in vitro tests on mice, and the results showed that 7-DGD alleviated mice mortality induced by LPS [97]. This substance improves inflammation by activating the Keap1/Nrf2/HO-1 signaling pathway. In addition, many natural substances isolated from *T. sinensis* have been reported to possess notable anti-inflammatory effects [98,99].

### 6.4. Anti-Cancer Effect

As a natural anti-cancer drug, extract of *T. sinensis* is attracting increasing attention. Zhang et al. (2014) extracted four compounds from *T. sinensis* leaves (quercetin-3-*O*-α-L-rhamnopyranoside, kaempferol-3-*O*-α-L-rhamnopyranoside, 1,2,3,4,6-penta-*O*-galloyl-*β*-D-glucopyranose and ethyl gallate), and reported that Kaempferol-3-*O*-α-L-rhamnopyranoside can inhibit the proliferation of HepG$_2$ human liver cancer cells and MCF-2 human breast cancer cells as well as induce apoptosis [100]. The leaves of *T. sinensis* are rich in gallic acid, an important anti-cancer substance that can promote DU145 prostate cell apoptosis through the production of reactive oxygen species and mitochondrial pathways [101] as well as induce the apoptosis of oral squamous cancer cells by up-regulating

the pro-apoptotic genes (*TNF-α*, *TP53BP2* and *GADD45A*) and down-regulating the anti-apoptotic genes (*Survivin* and *cIAP1*) [102]. In addition, betulonic acid and 3-oxours-12-en-28-oic acid are both found in *T. sinensis*, which block the proliferation of MGC-803 and PC3 cancer cells and induce their apoptosis through the mitochondrial p53, bax, caspase 9, and caspase 3 pathways [103].

*6.5. Hypoglycemic Effect*

For nearly twenty years studies on the hypoglycemic effects of *T. sinensis* extracts have been reported, which could be beneficial for diabetes patients. In 2003, Yang et al. (2013) reported that ethanol extracts of *T. sinensis* leaf could enhance cellular glucose uptake in basal and insulin-stimulated 3T3-L1 adipocytes [104]. Hsieh et al. (2005) reported the inhibitory effect of *T. sinensis* extracts on LDL glycation induced by glucose and glyoxal [105]. Two studies have indicated that flavonoids of *T. sinensis* might be the active constituents corresponding to the hypoglycemic effects of this plant [106,107]. Furthermore, studies on the extract of *T. sinensis* have revealed that the mechanisms of TSL stimulating glucose uptake and ameliorating insulin resistance might be related to AMPK activation in skeletal muscles and to up-regulation of PPARγ and normalized adiponectin in adipose tissues [108].

## 7. Research on Genes and Omics

*7.1. Function Genes*

*T. sinensis* is rich in lignin and anthocyanin, two substances that are important indicators of bud quality. Cinnamic alcohol-CoA reductase (CCR) and cinnamyl alcohol dehydrogenase (CAD) are the key enzymes in lignin biosynthesis, and chalcone isomerase (CHI) and anthocyanidin reductase (ANR) are required for plant anthocyanins biosynthesis. Based on the RNA-seq data of *T. sinensis*, *TsCCR* [109], *TsCCR1* [110], *TsCAD1* [111], *TsCHI* [112], *TsANR* [113] were identified and cloned. The *TsCCR* gene contained an open reading frame of 975 bp and encoded putative polypeptides of 324 amino acid residues [109], and the *TsCCR1* gene contained an open reading frame of 924 bp and encoded putative polypeptides of 307 amino acid residues [110]. The *TsCAD1* gene contained an open reading frame of 1,068 bp and encoded putative polypeptides of 355 amino acid residues [111]. The *TsCHI* gene contained an open reading frame of 717 bp and encoded putative polypeptides of 238 amino acids [112]. The *TsANR* gene contained an open reading frame of 1011 bp and encoded putative polypeptides of 336 amino acids [113].

Few studies have reported the expression patterns of the four genes (*TsCCR*, *TsCAD1*, *TsCHI*, and *TsANR*) in different tissues of the *T. sinensis* plant; the results showed that the expression level of *TsCCR* and *TsCHI* genes in stems were significantly higher than those in roots and leaves, and the transcript level of *TsCAD1* gene in roots was significantly higher than in the stems and leaves. The results of real-time PCR showed the highest relative expression of *TsANR* gene in the leaves of *T. sinensis* seedlings.

Moreover, the expression pattern of these four genes under different stress treatments were investigated, and the results showed that the expression pattern of the *TsCCR* gene was first temporarily up-regulated and then down-regulated after cold treatment, which is the opposite of the expression pattern of this gene during heat treatment. During a salt stress treatment (200 mmol·L$^{-1}$ NaCl), *TsCCR* expression decreased significantly in the first 4 h, then increased rapidly, decreased, increased, and then decreased again during 200 g·L$^{-1}$ PEG6000 drought stress treatment [109]. During 24 h 38 °C heat treatment, *TsCAD1* expression first decreased and then increased significantly, while it first increased and then decreased during 24 h 4 °C cold treatment. *TsCAD1* expression first decreased, then increased, and then decreased again during 24 h drought stress treatment (200 g·L$^{-1}$ PEG 6000), although the relative expression level was lower than control. The expression trend during a 200 mmol·L$^{-1}$ NaCl salt stress treatment for 24 h first decreased and then increased, although the relative expression level was lower than the control [111]. Under high temperature (38 °C), drought stress (200 g·L$^{-1}$ PEG 6000 solution) and salt stress (200 mmol·L$^{-1}$ NaCl solution), *TsCHI* expression was higher than that of the control, and

showed an upward trend as the processing time was extended [112]. These results provide a theoretical basis for genetically engineering *T. sinensis* to cultivate resistances against extreme environmental conditions. With 24 h of high temperature treatment (38 °C), *TsANR* gene expression increased significantly at 1 h, then returned to the level of control [113].

*7.2. Omics*

*T. sinensis* omics research has mainly focused on genomic and transcriptomic studies. Ran et al. (2020) sequenced the transcriptomes of the buds of the *T. sinensis* varieties 'Heiyouchun' and 'Qingyouchun' during four developmental periods, then analyzed the expression pattern of anthocyanin biosynthesis genes. Among the key genes expressed in anthocyanin synthesis by KEGG analysis, five genes, namely, phenylalanine ammonia lyase (PAL), coumarin-Coa ligase (4CL), Chalketone synthase (CHS), flavonoid 3-hydroxylase (F3'H), and anthocyanin synthase (ANS), were up-regulated in 'Heiyouchun', while C3'H and flavonol synthase (FLS) were down-regulated in 'Heiyouchun' [114]. Zhao et al. (2017) analyzed the RNA-seq data of 'Heiyouchun' sprouts and found 467 unigenes involved in terpenoid biosynthesis related to flavor formation, including 226, 71, 86, and 84 unigenes for terpenoid backbone, monoterpenoid, sesquiterpenoid (triterpenoid), and diterpenoid biosynthesis, respectively [115]. Sui et al. (2019) analyzed the RNA-seq data of young leaves and mature leaves of *T. sinensis,* and found that the KEGG pathways for phenylpropanoid, naringenin, lignin, cutin, suberin, and wax biosynthesis were significantly enriched in mature leaves [116].

Xiang et al. (2021) assembled the complete *T. sinensis* chloroplast genome using second-generation high-throughput sequencing technology. The chloroplast genome contained 138 genes in total, including 89 protein-coding genes, seven rRNA genes, forty tRNA genes, and two pseudogenes [117]. Liu et al. (2019) sequenced the chloroplast genome of *T. sinensis* using an Ilumina sequencing platform, and found that the chloroplast genome is a characteristic four-party structure with a length of 157,228 bp which contains two 26,994 bp inverted repeats (IRs), an 85,971 bp large single-copy, and a 17,269 bp small single-copy. A total of 126 genes, including 82 protein-coding genes, 36 tRNA genes, and eight rRNA genes, were identified [118]. Ji et al. (2021) reported a high-quality *T. sinensis* genome assembly with scaffolds anchored to 28 chromosomes, an assembled length of 596 Mb, and a total of 34,345 genes predicted in the genome after homology-based and de novo annotation analyses [119].

**8. Conclusions**

This paper comprehensively presents the storage, preservation, processing, nutrient compounds, chemical compounds, phytochemistry, function genes, and omics of *T. sinensis*. There are three kinds of storage and preservation methods of *T. sinensis* buds; although they can extend the shelf life of *T. sinensis*, the nutrients in this vegetable plant experience different degrees of loss. Because bio-preservation methods have the advantages of being natural, safe, and simple, it has become one of the research hotspots of food preservation technology. However, there has been less research on bio-preservation methods for *T. sinensis*. Furthermore, in order to extend the consumption period of *T. sinensis*, it is processed into a wide variety of foods.

Additionally, in the present review, volatile, terpenoids, phenylpropanoids, and flavonoids from different parts of this plant were summarized; the existing pharmacological investigations have revealed that this plant have a wide spectrum of pharmacological effects, in particular for its anti-cancer and anti-inflammatory activities. However, a large amount of pharmacological activity is associated with gallic acid, and the pharmacological activity of other bioactive substances has been less studied. Therefore, the pharmacological effects of other chemical components and the drug development of *T. sinensis* extracts need a great deal of additional detailed research.

At present, research on genes of *T. sinensis* mainly focuses on two aspects, namely, anthocyanin-related genes and lignin-related genes. In the storage process, *T. sinensis* is

prone to anthocyanin degradation and lignin accumulation, which affects the edible taste of *T. sinensis* buds. By studying anthocyanin-related genes and lignin-related genes, the problems of anthocyanin degradation and lignin accumulation during the shelf life of *T. sinensis* buds can be further solved by genetic means. This basic research can provide a certain theoretical basis for germplasm innovation with *T. sinensis*.

In conclusion, this paper highlights the importance of this plant and provides direction for future food and drug development and germplasm innovation with *T. sinensis* by providing detailed information *T. sinensis* as a plant diversity resource.

**Author Contributions:** Conceptualization, Q.Z., P.-H.M. and G.-F.T., J.Z.; methodology, Q.Z., G.-F.T., X.-L.Z. and S.-H.Z.; data curation, Q.Z., S.-H.Z., K.W.; writing—original draft preparation, Q.Z. and G.-F.T.; writing—review and editing, Q.Z., G.-F.T., X.-L.Z., S.-H.Z., K.W., visualization, Q.Z.; funding acquisition, P.-H.M., G.-F.T. and J.Z. All authors have read and agreed to the published version of the manuscript.

**Funding:** This research was funded by the Project of Guizhou Academy of Agricultural Sciences (Support of Guizhou Academy of Agricultural Sciences No. [2021] 05, Germplasm Resources of Guizhou Academy of Agricultural Sciences No. [2020] 10); Jilin Agricultural University High Level Research Grant (JAUHLRG20102006); High Level Innovative Talents Training, Hundred Level Talents Project (Qiankehe talent [2015] 4024).

**Institutional Review Board Statement:** Not applicable.

**Data Availability Statement:** Not applicable.

**Acknowledgments:** Data can be found within the manuscript.

**Conflicts of Interest:** There is no conflict of interest.

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
