# Peer review of "Research Advances in Toona sinensis, a Traditional Chinese Medicinal Plant and Popular Vegetable in China"

_diversity, doi:10.3390/d14070572_

Round 1

Reviewer 1 Report

Dear authors,  After reviewing the following manuscript entitled "Research advances in Toona sinensis, a traditional chinese medicinal plant and notable vegetable" and with reference number (Diversity - 1811594), I sent the following comments and observations that the authors should attend to before its publication in this journal.   I appreciate the work being well organized, with the experimental data explained. However, I have a few remarks:  

It is good that in the title of the manuscript the binomial name of the plant is written correctly

Being an approach to the plant as a product with pharmacological benefits, I recommend eliminating notable vegetables”.

Chapter 3 - a study from 1994 and 1997 is mentioned. Information from more recent articles would also be welcome.

Author Response

diversity-1811594

Reviewer #1:

Dear authors,  After reviewing the following manuscript entitled "Research advances in Toona sinensis, a traditional chinese medicinal plant and notable vegetable" and with reference number (Diversity - 1811594), I sent the following comments and observations that the authors should attend to before its publication in this journal.   I appreciate the work being well organized, with the experimental data explained. However, I have a few remarks:  

It is good that in the title of the manuscript the binomial name of the plant is written correctly

Being an approach to the plant as a product with pharmacological benefits, I recommend eliminating “notable vegetables”.

Chapter 3 - a study from 1994 and 1997 is mentioned. Information from more recent articles would also be welcome.

Response: 

We thank the reviewer for the valuable observation and suggestion.

  1. We had omitted “notable vegetable” from the title.
  2. Chapter 3- a study from 1994 and 1997 is mentioned. Information from more recent articles would also be welcome.

We have added more current relevant reference to manuscript

Once again, we thank the reviewer for the critics and suggestion.

Reviewer 2 Report

Toona sinensis is an interesting plant originated in the north part of China. Review about pharmacological value, chemicals identified and food and vegetable have of much value. But little information about processing of food and vegetables. Please explain more about food use. In the title, "vegetable" the readers are anticipating about vegetable use of this plant and their value in connection with medicinal value.

About amino acids, better to rearrange the order of amino acid, from essential amino acid and others and should be consistent.  As for chemicals, should be focus on the chemicals specific in this plant, such as toonin and newly identified compounds. For these focused chemicals, should show chemical structure. I hope the authors re-wright this review.  

Author Response

Diversity-1811594

Reviewer #2:

Toona sinensis is an interesting plant originated in the north part of China. Review about pharmacological value, chemicals identified and food and vegetable have of much value. But little information about processing of food and vegetables. Please explain more about food use. In the title, "vegetable" the readers are anticipating about vegetable use of this plant and their value in connection with medicinal value.

About amino acids, better to rearrange the order of amino acid, from essential amino acid and others and should be consistent.  As for chemicals, should be focus on the chemicals specific in this plant, such as toonin and newly identified compounds. For these focused chemicals, should show chemical structure. I hope the authors re-wright this review.  

Response: 

We thank the reviewer for the critics and valuable suggestion.

  • But little information about processing of food and vegetables. Please explain more about food use. In the title, "vegetable" the readers are anticipating about vegetable use of this plant and their value in connection with medicinal value.

Re: We thank the reviewer’s commons and suggestions. We have added information Toona sinensisas vegetable and food processing methods which are highlighted in red line 118-125.

  1. About amino acids, better to rearrange the order of amino acid, from essential amino acid and others and should be consistent.  As for chemicals, should be focus on the chemicals specific in this plant, such as toonin and newly identified compounds. For these focused chemicals, should show chemical structure. I hope the authors re-wright this reviewWe have added more current relevant reference to manuscript

Re: Once again, we thank the reviewer for the critics and suggestion.

  • Essential amino acids from Toonasinensis’sdifferent tissue and developmental status are listed consistent.
  • The chemical structure like toonin has been added in the revised manuscript.
  1. All the commons and critical suggestions from reviewer’s PDF file have been responded and dealt in red
  • Better to mention which part (fruit, leaf, etc) is difficult for storage.

Re: We thank the reviewer’s commons and suggestions. We have added T. sinensiswhich parts are difficult to store in red line 17.

  • In this review, mostly pharmacological review, almost no information about food...There are already seveal review about this plant for chemicals and medicinal use. But there are little about food and medicinal use. Please explain more about food use. In the title, "vegetable" the readers are anticipating about vegetable use of this pland and their value.

Re: We thank the reviewer’s commons and suggestions. We have addthe vegetable use of T. sinensis buds in Chapter 3.1.

  • Please explain more about how to eat. Only this picture, difficult to understand.

Re: We thank the reviewer’s commons and suggestions. We have addthe edible methods of T. sinensis buds in Chapter 3.1.

  • Please explain more about how to eat. Only this picture, difficult to understand.

Re: We thank the reviewer’s commons and suggestions. We have addthe explanation of the picture in red line 55.

  • Benzyl adeine? Better to show full name.

Re: We thank the reviewer’s commons and suggestions. The full name of 6-BA is 6-Benzylaminopurine.

  • I anamticipating to explain more about these processing.

Re: We thank the reviewer’s commons and suggestions. We have added the more explain of theseprocessing in red line 133-134.

  • Essential amino acid should be consistent to other Tables.

Re: We thank the reviewer’s commons and suggestions. We have revisedthe essential amino acid of the tables. And the modifications are shown in Tables 2 and 3.

  • Why two stearic acid?

Re: We thank the reviewer’s commons and suggestions. We have revisedthe two stearic acid. And the content as follow: 1.88% stearic, 3.44% oleic acid in red line 147-148.

  • Order of amino acids should be consistent, better to separete "essential amino acid " and other amino acids.

Re: We thank the reviewer’s commons and suggestions. We have revisedthe content in Table 2 and Table 3. The amion acid in Table 2 and Table 3 are consistent, and the essential amino acids arelabeled with *, which distinguishes other amino acids.

  • Should be consistent to Table 3.

Re: We thank the reviewer’s commons and suggestions. We have revisedthe Table 3. And the Table 2 and Table 3 are consistent.

  • Better to show the structure of chemicals specific in this plant, such as toonin C,

Re: We thank the reviewer’s commons and suggestions. We have addedthe structureof toonin Cin figure 2.

  • Methyl gallate? Common to many palnts, not specific to sinensis.

Re: We thank the reviewer’s commons and suggestions. We have revised methyl gallate isolated from Toona sinensisin red line 256.

  • These Japnese name are given name. Should be changed to family name, the same as others

Re: We thank the reviewer’s commons and suggestions. We have revised this reference. Amend as follows:

Kakumu, A.; Ninomiya, M.; Efdi, M.; Adfa, M.; Hayashi, M.; Tanaka, K.; Koketsu, M. Phytochemical analysis and antileukemic activity of polyphenolic constituents of Toona sinensis. Med. Chem. Lett. 2014, 24, 4286-4290.

Once again, we thank the reviewer for the critics and suggestion.

Round 2

Reviewer 2 Report

The authors revised according to my suggestions. Now I agree this review will be published.